# Alternative Pathway is Involved in Nitric Oxide-Enhanced Tolerance to Cadmium Stress in Barley Roots

**DOI:** 10.3390/plants8120557

**Published:** 2019-11-29

**Authors:** Li He, Xiaomin Wang, Ruijun Feng, Qiang He, Shengwang Wang, Cuifang Liang, Lili Yan, Yurong Bi

**Affiliations:** 1Ministry of Education Key Laboratory of Cell Activities and Stress Adaptations, School of Life Sciences, Lanzhou University, Lanzhou, Gansu 730000, China; hel16@lzu.edu.cn (L.H.); wangxiaomin@lzu.edu.cn (X.W.); fengrj16@lzu.edu.cn (R.F.); heq15@lzu.edu.cn (Q.H.); wangshw16@lzu.edu.cn (S.W.); liangcf16@lzu.edu.cn (C.L.); yanll18@lzu.edu.cn (L.Y.); 2State Key Laboratory of Plateau Ecology and Agriculture, Qinghai University, Xining, Qinghai 810016, China

**Keywords:** alternative pathway, Cd stress, nitric oxide, reactive oxygen species

## Abstract

Alternative pathway (AP) has been widely accepted to be involved in enhancing tolerance to various environmental stresses. In this study, the role of AP in response to cadmium (Cd) stress in two barley varieties, highland barley (Kunlun14) and barley (Ganpi6), was investigated. Results showed that the malondialdehyde (MDA) content and electrolyte leakage (EL) level under Cd stress increased in two barley varieties. The expressions of alternative oxidase (*AOX*) genes (mainly *AOX1a*), AP capacity (V_alt_), and AOX protein amount were clearly induced more in Kunlun14 under Cd stress, and these parameters were further enhanced by applying sodium nitroprussid (SNP, a NO donor). Moreover, H_2_O_2_ and O_2_^−^ contents were raised in the Cd-treated roots of two barley varieties, but they were markedly relieved by exogenous SNP. However, this mitigating effect was aggravated by salicylhydroxamic acid (SHAM, an AOX inhibitor), suggesting that AP contributes to NO-enhanced Cd stress tolerance. Further study demonstrated that the effect of SHAM application on reactive oxygen species (ROS)-related scavenging enzymes and antioxidants was minimal. These observations showed that AP exerts an indispensable function in NO-enhanced Cd stress tolerance in two barley varieties. AP was mainly responsible for regulating the ROS accumulation to maintain the homeostasis of redox state.

## 1. Introduction

Cadmium (Cd), one of the most toxic heavy metals to the environment, has drawn great attention worldwide. It not only inhibits plant growth, but also pollutes the food chain, seriously threatening human health [1,2]. When Cd is accumulated excessively, at the physiological level, a suite of symptoms, such as chlorosis, underdevelopment, and programmed cell death, are induced in plants [3,4,5]. At the cellular level, over-accumulated Cd affects enzyme activity and changes protein structure [4,6,7]. More importantly, cellular redox homoeostasis is disturbed, and reactive oxygen species (ROS) burst, which further leads to oxidative stresses. To better deal with Cd toxicity, plants have evolved various defense strategies. Activating antioxidant enzymes and non-enzymatic antioxidants to counteract the oxidative stress has been widely accepted [8,9]. Alternative pathway (AP) also contributes to enhancing Cd tolerance [10]. Even though great effort has been made in the past, the protective mechanisms against Cd stress still need to be explored.

Nitric oxide (NO), known as a multifunctional signaling molecule, has been widely reported upon in recent years. It not only functions in plant physiological processes, but also defends against various environmental stresses [11,12,13], even mitigating heavy metal toxicity [14,15]. There are enzymatic and non-enzymatic systems that produce NO [16,17]. Increasing findings on defensive roles of NO to Cd toxicity have been widely reported. At the cellular level, to reduce Cd-induced ROS over-accumulation, NO enhances the antioxidant enzyme defense system among others [8,18,19,20]. At the molecular level, NO, as a signaling molecule, enhances Cd stress response by cross-talking with auxin [21,22], abscisic acid [23], salicylic acid [24], polyamine [25], Ca^2+^ [25], and hydrogen sulfide [26], or by regulating protein kinase activities [27]. Although numerous studies have shown that NO could enhance Cd stress tolerance in different plants, there is little evidence to illuminate whether exogenous sodium nitroprussid (SNP; a NO donor) can improve Cd tolerance in highland barley.

Respiration has momentous functions in plant cell metabolism. Plant mitochondria possess a cyanide-resistant AP apart from the cyanide-sensitive cytochrome pathway (CP) [28]. Alternative oxidase (AOX), oriented in the mitochondria inner membrane, is a terminal oxidase of AP. When electrons flow to AOX, ROS production will markedly decrease, which helps hamper oxidative damages. When plants are exposed to low nitrogen stress, drought, and low temperature stresses, AP activity is obviously elevated [29,30,31]. Some findings have deemed that AP contributes to improving tolerance to various environmental stresses [32,33,34]. As outlined recently, NO can enhance AP to improve plant tolerance [35,36]. In consideration of lacking NO-related mutants, SNP has been widely used to mimic endogenous NO in the study of its various biological functions in plants [21,37]. Applying SNP upregulates the expression level of the *AOX1a* gene and the AOX activity in *Arabidopsis* [38]. NO is in charge of the expression activation of *AOXla* in tobacco [39]. Virus infection induces NO production, which might function as an upstream signal molecule to induce AP [35,36]. However, there is little evidence to confirm whether AP contributes to SNP-elevated tolerance to Cd stress in highland barley.

Highland barley, also known as naked barley, has been regarded as an excellent material to initiate mechanism study of crop tolerance to various stresses, as it generally grows in regions more than 3000 m above sea level where climate conditions are complicated and severe, including low temperature, high light intensity, and desolate soil, among other factors. In this study, three members (*AOX1a*, *AOX1d1,* and *AOX1d2*) of the *AOX* family were identified in highland barley. So far, there is little research that has studied the function of AOX in highland barley. In this study, we examined the response of AP to Cd stress in highland barley. Meanwhile, the function of NO in mediating AP under Cd stress was investigated. Moreover, the source of endogenous NO production was further clarified. The results indicated that AP plays an indispensable role in the SNP-enhanced Cd tolerance in highland barley by regulating the accumulation of ROS.

## 2. Materials and Methods

### 2.1. Plant Materials and Growth Conditions

Barley varieties Ganpi6 and Kunlun14 were used in the present study, and Ganpi6 was regarded as the control. Seeds were surface-disinfected with 2% hypochlorite for 8 min, and washed thoroughly with sterile water. Then, seeds were transferred onto a nylon net on top of 200 mL plastic beakers (20 seedlings per beaker), which were filled with 1/4-strengh modified culture solution [40]. Materials were allowed to grow in a chamber with 16 h light/8 h dark cycles. Culture solution was changed every 2 days. Different chemicals were added to the medium for various treatments after 6 days. Roots were utilized immediately for the following assays.

### 2.2. Root Elongation Measurement

After 6 day growth, different concentrations of Cd were added to 200 mL 1/4-strengh modified Johnsons nutrient solution for 48 h [40]. The roots were collected immediately for measurement of root length with Image J.

### 2.3. Root Electrolyte Leakage Determination

Electrolyte leakage (EL) was monitored according to the method described by Tang et al. (2014) [41]. First, 0.2 g roots were washed in the deionized water for more than three times. The roots were dipped in the deionized water for 2 h at 25 °C. The conductivity of the water bath (C_1_) and the de-ionized water (C_0_) was measured. Root samples were then boiled at 100 °C for 40 min, and the conductivity of the water bath (C_2_) was measured when it cooled to room temperature.

### 2.4. Malonaldehyde Content Determination

Malondialdehyde (MDA) content was determined according to the methods described by Tang et al. (2014) [41] and Wang et al. (2012) [42]. Grinding 0.3 g roots in 2 mL 10% trichloroacetic acid (TCA) and centrifuging at 10,000× *g* for 10 min, the supernatant was then incubated with 1 mL TCA (0.5%) at 95 °C for 30 min, and centrifuged at 10,000× *g* for 10 min. The absorbance value was read at 532, 440, and 600 nm.

### 2.5. Measurements of Respiration Rate

Respiration was measured on the basis of the method described by Wang et al. [43]. First, 0.03 g roots were cut into 2 mm small segments and put into 2 mL 50 mM phosphate buffer (pH 6.8). After reaction for 2 min at room temperature, the oxygen value slope was defined as the total respiration rate (V_t_). Then, 2 mM KCN or 2 mM salicylhydroxamic acid (SHAM) was added and reacted for 2 min; the oxygen value slope was defined as the AP capacity (V_alt_) or the CP capacity (V_cyt_), independently.

### 2.6. Determination of NO Content

NO level was analyzed on the basis of the method described by Wang et al. [19]. Roots were bathed in the buffer containing [20 mM 4-amino-5-methylamino-2′7′-difluorofluorescein diacetate (DAF-FM-DA, NO fluorescent probe), 0.25 mM NaCl, 1 mM CaCl_2_, 10 mM Hepes-NaOH (pH 7.0)] for 30 min. Then, roots were washed more than three times with 10 mM Hepes-NaOH (pH 7.0)] buffer, and scanned in the Leica SM IRBE stereomicroscope. The images were analyzed by the Leica ZEN software.

### 2.7. Determination of Nitrate Reductase (NR) Activity

NR activity was measured following the method described by Mackintosh et al. [44]. First, 0.5 g roots were extracted in extract solution A containing 50 mM Hepes-KOH (pH 7.5), 5% glyceol, 1 mM PMSF (Phenylmethanesulfonyl fluoride), 1 mM dithiothreitol (DTT), 10 mM MgCl_2_, and 10 μM FAD (flavin adenine dinucleotide). After being centrifuged at 15,000× *g* for 25 min, 250 μL of supernatant was added to 250 μL of reaction solution Hepes-KOH (pH 7.5, 50 mM), KNO_3_ (2 mM), NADH (reduced nicotinamide-adenine dinucleotid) (0.2 mM), DTT (1 mM), MgCl_2_ (10 mM)) and incubated for 15 min. A total of 50 μL of 0.5 M zinc acetate was used to terminate the reaction. Following addition of 150 μL of 1% sulfanilamide and 150 μL 0.02% (w/v) *N*-(1-naphthy1) ethylenediamine dihydrochloride, the absorbance at 540 nm was recorded.

### 2.8. Determination of Nitric Oxide Synthase (NOS) Activity

NOS activity was measured according to the method described by Lin et al. [45]. First, 0.25 g roots were homogenized with 2 mL of extraction buffer and centrifuged at 15,000× *g* for 15 min. Then, 100 μL of supernatant was added to the reaction solution, and 0.2 mM NADPH (reduced nicotinamide adenine dinucleotide phosphate) was added to initiate the reaction. The absorbance at 340 nm was read.

### 2.9. Cloning, Sequencing, and Bioinformatics Analyses

To obtain the intact open reading frame of *HvAOX1*, primers (Table 1) were designed on the basis of *AOX* sequence in highland barley roots. Genbank accession numbers were AK363239.1 (*HvAOX1a*), AK365405.1 (*HvAOX1d1*), and AK251266.1 (*HvAOX1d2*). A total of 15 μL reaction mixture was used for PCR. The PCR conditions were 2 min at 98 °C, then 40 cycles of 20 sec at 98 °C, 30 sec at 56.0–58.5 °C, 10 min at 72 °C, and 10 min at 25 °C. The clear single target band was cloned to the pBlunt vector (TransGen Biotech, Beijing, China).

### 2.10. RNA Isolation and qRT-PCR

About 100 mg samples were used for total RNA isolation from various plant tissues by using Trizol reagent. [cDNA synthesis kit (TRANS, Beijing, China)] was used for cDNA synthesis. The qRT-PCR mixture contained 5 μL SYBR Green I Master Mix, 0.5 μL forward primers, 0.5 μL reverse primers, and 4 μL deionized water. Each gene was amplified in three biological replicates. The results were analyzed by Rotor-Gene Real-Time Analysis Software 6.1. The specific primers are listed in Table 1.

### 2.11. Western Blot Analysis

Western blot analysis was carried out following the method described by Zhao et al. [46]. Proteins were separated on a 10% acrylamide gel. After electrophoresis, the proteins were transferred to a polyvinylidene difluoride membrane. The membrane was blocked for 3 h with 10% bovine serum albumin in buffer solution (150 mM NaCl, 0.05% Tween-20, 10 mM Tris-HCl (pH8.0)). Primary antibody against *Arabidopsis* AOX was added and incubated for 10 h. After rinsing three times, secondary antibody was added and incubated for visualization.

### 2.12. H_2_O_2_ and O_2_^−^ Staining

H_2_O_2_ and O_2_^−^ staining was carried out according to the method described by Wang et al. [47]. Roots were stained in 0.5 mg/mL NBT (nitro blue tetrazolium) solution for 2 h or 2 mg/mL DAB (3,3-diaminobenzidine) solution for 12 h, then de-stained in 95% ethanol for 40 min.

### 2.13. Extraction and Estimation of Antioxidants

First, 0.25 g fresh roots were ground in 2 mL meta-phosphoric acid (5%) and centrifuged at 11,000× *g* for 20 min, and then 1.0 U ascorbate oxidase was added. The absorbance was read at 265 nm. Oxidized ascorbate (DHA, dehydroascorbic acid) content was equal to the total ascorbic acid (AsA) content minus reduced AsA. Oxidized glutathione (GSSG) and reduced glutathione (GSH) contents were measured according to the method described by Giraud et al. [48].

### 2.14. Antioxidant Enzyme Activity Assay

First, 0.25 g fresh roots were ground in 3 mL pre-cooled 25 mM Hepes buffers, and then were centrifuged at 4 °C for 20 min at 12,000× *g*. The supernatants were collected for examining the activity of antioxidant enzymes. SOD (Superoxide dismutase), CAT (Catalase), POD (Peroxidase) and APX (Ascorbate peroxidase) activities were measured on the basis of the method described by Jian et al. [36]. DHAR (Didehydroascorbic acid reductase), MDHAR (Monodehydroascorbic acid reductase), GR (Glutathione reductase), and GPX (Glutathione peroxidase) activities were measured following the method described by Zhang et al. [49].

### 2.15. Statistical Analysis

Each experiment was repeated at least three times with three replicates. The confidence coefficient was set at 0.05. Experiments that required an analysis of variance were analyzed using SPSS 17.0 analysis of variance (ANOVA) and Origin 8.

## 3. Results

### 3.1. Effects of Cd Stress on Dry Weight, Root Elongation, MDA Content, and EL

Cd stress led to toxicity symptoms and inhibited the elongation of Ganpi6 and Kunlun14 roots in a dose-dependent manner. After exposure to various concentrations (0–200 μM) of Cd, the root growth of barley (Ganpi6) and highland barley (Kunlun14) was gradually inhibited (Figure 1A,B). Under 150 μM Cd treatment, dry weight in Ganpi6 and Kunlun14 plants was reduced by 42.29% and 33.49%, respectively (Figure 1C), whereas root elongation in Ganpi6 and Kunlun14 was decreased by 55.2% and 50.9%, respectively (Figure 1D). To further explore the cellular membrane damage caused by Cd stress, two important indicators, MDA and EL levels, were examined. As shown in Figure 1E,F, the MDA content in Ganpi6 and Kunlun14 roots in the presence of 150 μM Cd was elevated 5.83-fold and 4.69-fold, respectively, whereas the EL was increased by 80.3% and 70.7%, respectively. Thus, 150 μM Cd was selected for the subsequent assays.

### 3.2. Effects of Cd Stress on the Respiratory Pathways

To explore the response of respiration to Cd stress, V_t_, V_cyt_, and V_alt_ in Ganpi6 and Kunlun14 were examined under various Cd concentrations. As shown in Figure 2A, V_t_ was rapidly increased when Cd concentration was lower than 150 μM, then it showed a declined trend. Compared with V_cyt_, V_alt_ showed a similar pattern observed in V_t_ under Cd stress (Figure 2B,C). It was noteworthy that V_alt_, V_alt_/V_cyt_ were higher in Kunlun14 than those in Ganpi6 (Figure 2), suggesting that V_alt_ in Kunlun14 had a greater contribution to V_t_ than Ganpi6 did under Cd stress. The time course of respiration changes under 150 µM Cd treatment were examined. The results showed that V_t_, V_alt_, and V_cyt_ in both Ganpi6 and Kunlun14 reached their maximal values at 12 h, followed by decreases. Nevertheless, all of them were still markedly higher than that in control roots (Figure 2E–G).

### 3.3. Cd Stress Induced NO Production from NR Pathway

More and more evidence indicates that NO is involved in enhancing plant tolerance to Cd [50,51]. To explore whether Cd stress excites NO release in Ganpi6 and Kunlun14, NO production was imaged by using DAF-FM, a fluorescent probe, together with NO donor (SNP) or scavenger (PTIO) treatment. In Ganpi6 and Kunlun14 roots, weak NO fluorescence signal was observed in control roots. However, Cd stress greatly enhanced the fluorescence signal, which was further intensified by SNP treatment (Figure 3A,B). Notably, the NO fluorescence signal was stronger in Kunlun14 than that in Ganpi6. Applying PTIO almost eliminated Cd-induced NO fluorescence signal (Figure 3A,B). To further explore the source of Cd-induced endogenous NO production in Ganpi6 and Kunlun14 roots, NR inhibitor (NaN_3_) and NOS inhibitor (L-NNA) were used to detect NO original level. The results showed that NaN_3_ in Ganpi6 and Kunlun14 roots nearly abolished Cd-induced NO accumulation, whereas applying L-NNA had almost no impact on Cd-induced NO accumulation (Figure 3A), indicating that the Cd-induced NO production originated from the NR-dependent pathway. To further confirm this conclusion, NR and NOS activities were examined. Under Cd stress, NR activity in Ganpi6 and Kunlun14 roots was increased by 57.6% and 72.1%, respectively (Figure 3C). Furthermore, the NR activity showed a similar pattern as that of NO production (Figure 3B,C). However, the NOS activity had little change under Cd stress (Figure 3D).

Some studies reported that mitochondria might be an important source for NO generation via single electron leak from the electron transfer chain (ETC) to nitrite [52,53]. Moreover, recent studies have confirmed that AOX functions to protect ETC components from over-reduction in plants, thus preventing single electron leak [54]. To explore the link between NO and AP under Cd stress in Ganpi6 and Kunlun14 roots, SHAM (AOX inhibitor) was used to block the function of AP. As shown in Figure 3, under Cd + SHAM stress, NO fluorescence signal was strongly enhanced, suggesting that dysfunctional AP leads to over-reduction of ETC components.

### 3.4. Expression Patterns of AOX Genes in Highland Barley

cDNA of *AOX* was cloned from highland barley using the candidate gene approach [55]. Comparison with the nucleotide sequences to barley *AOXs* confirmed that the highland barley *AOXs* sequences were *AOX1* genes. These genes were named *HvAOX1a*, *HvAOX1d1*, and *HvAOX1d2*. The corresponding accession numbers were MK361118, MK361119, and MK361120. The transcript levels of *HvAOX* genes in different developmental stages of highland barley were analyzed using qRT-PCR. As shown in Figure 4, three *HvAOX* genes were detected in all tissues but with specific expression patterns. The expression of *HvAOX1a* was higher than that of *HvAOX1d1* and *HvAOX1d2* in all tissues except in stamens. It was worth mentioning that the transcript levels of *HvAOXs* showed similar increasing pattern with the leaf age (from first leaves to function leaves), implying a common developmental-related feature of all *HvAOX* genes (Figure 4).

### 3.5. Exogenous NO Enhanced HvAOX Expression in Ganpi6 and Kunlun14 Roots under Cd Stress

AP usually runs at a low level, however, it could be observably induced when plants suffer from various environmental stresses [25]. To examine whether the increase of V_alt_ in Cd-stressed roots is regulated at the transcriptional level, *HvAOX* expression was investigated. As shown in Figure 5, *HvAOX1a*, *HvAOX1d1*, and *HvAOX1d2* in Ganpi6 and Kunlun14 were all expressed in roots. Compared with *HvAOX1d1* and *HvAOX1d2,* the expression level of *HvAOX1a* was markedly increased under Cd stress. To further explore how the expression levels of *HvAOXs* were regulated in Cd-stressed roots, we manipulated the production of NO and monitored *HvAOXs* expression. Under Cd + SNP treatment, the expression of *HvAOX1a* was increased by 3.5 times and 4.4 times in Ganpi6 and Kunlun14 roots, respectively (Figure 5A), whereas *HvAOX1d1* and *HvAOX1d2* were just slightly increased (Figure 5B,C). Under Cd + SHAM treatment, the expression level of *HvAOXs* was dramatically reduced.

### 3.6. Exogenous NO Enhanced V_alt_ and AOX Protein Level under Cd Stress

To further explore NO effects on respiration under Cd stress, V_alt_ and V_cyt_ were examined. As shown in Figure 6A, under Cd + SNP treatment, V_alt_ in Ganpi6 and Kunlun14 roots was increased by 52.4% and 60.4%, respectively, whereas V_cyt_ had nearly no change. When AP was inhibited by SHAM under Cd stress, V_alt_ was decreased to nearly the control level, and V_cyt_ still had little change (Figure 6B). It is worth mentioning that when AP was blocked under Cd stress, applying SNP did not reverse the trend. To further decipher whether the increase of AP in Cd-stressed roots occurs at the protein level, the AOX protein level was detected. Western blotting results showed that Cd stress excited the AOX protein expression in two barley varieties, and the expression of AOX protein was higher in Kunlun14 than in Ganpi6. SNP treatment further enhanced the AOX protein levels in both Ganpi6 and Kunlun14 under Cd stress. However, AOX protein amount was markedly reduced when AP was inhibited by SHAM under Cd stress, and even exogenous SNP did not recover the reduction (Figure 6C).

### 3.7. Exogenous NO Did Not Relieve Cd-Induced Oxidative Stress under SHAM Treatment

To explore whether AP is engaged in Cd-induced oxidative damage in two barley varieties, MDA and EL levels were examined. The MDA content in Ganpi6 and Kunlun14 was increased by 78.7% and 62.2%, respectively, whereas the EL level was increased by 60.7% and 50.2%, respectively. Applying SNP greatly relieved the oxidative damage caused by Cd. The MDA content was decreased by 60.3% and 57.2% in Ganpi6 and Kunlun14, respectively, whereas EL was decreased by 33.4% and 30.7%, respectively (Figure 7C,D). Under Cd + SHAM treatment, the MDA content was increased by 38.6% and 33.5% in Ganpi6 and Kunlun14, respectively, whereas EL was increased by 29.5% and 26.3%, respectively, and the increases of MDA and EL were not reversed by SNP (Figure 7C,D). Because Cd-induced oxidative damage might be triggered by ROS accumulation, H_2_O_2_ and O_2_^−^ levels were examined by histochemical staining. As shown in Figure 7A,B, H_2_O_2_ staining was increased by 3.6 times and 2.8 times, and O_2_^−^ staining was increased by 3.2 times and 2.1 times in Ganpi6 and Kunlun14 roots under Cd stress, respectively. Applying SHAM further increased the H_2_O_2_ and O_2_^−^ staining under Cd stress, which was not attenuated by SNP treatment. The above results suggest that AP plays an indispensable role in NO relief of Cd-induced oxidative stress.

### 3.8. Effects of Exogenous NO on AsA and GSH Levels in the Presence of SHAM under Cd Stress

To elucidate the role of AP in controlling ROS homeostasis, the levels of small antioxidant molecules were examined. In Ganpi6 and Kunlun14 roots, an increment in the AsA level (0.8 times and 1.3 times, respectively), a decrease in the DHA level (15.2% and 15.5%, respectively), and an increase in the AsA/DHA ratio (1.2 times and 1.6 times, respectively) were observed under Cd stress. Applying SNP under Cd stress elevated the AsA level by 23.0% and 28.2%, and AsA/DHA ratio by 11.8% and 39.4% in Ganpi6 and Kunlun14 roots, respectively. However, the presence of SHAM, Cd, or Cd + SNP treatment had almost no impact on AsA or DHA levels in Ganpi6 and Kunlun14 roots (Figure 8A,B). The GSH level was elevated by 37.9% and 49.6% in Ganpi6 and Kunlun14 roots, respectively, whereas the GSSG level was decreased by 13.3% and 12.9%, respectively, and the GSH/GSSG ratio was increased by 49.6% and 72.0%, respectively. Applying SNP under Cd stress enhanced the GSH level by 10.9% and 12.1%, lowered the GSSG level by 33.3% and 38.7%, and elevated the GSH/GSSG ratio by 47.6% and 54.0% in Ganpi6 and Kunlun14 roots, respectively. Under Cd + SNP treatment, applying SHAM almost had no effects on GSH or GSSG levels in Ganpi6 and Kunlun14 roots (Figure 8D,E).

To further explore the mechanism of redox balance maintenance, activities of ascorbate-glutathione cycle-related enzymes were examined. Under Cd stress, the activities of GR was increased by 78.6% and 86.8%, MDHAR was increased by 28.9% and 34.3%, GPX was increased by 61.3% and 64.3%, and DHAR was increased by 38.4% and 39.7% in Ganpi6 and Kunlun14 roots, respectively. Under Cd + SNP treatment, the activities of GR and MDHAR were further enhanced, whereas the activities of GPX and DHAR were reduced in Ganpi6 and Kunlun14 roots. Under Cd + SNP + SHAM treatment, the activities of these enzymes were nearly undisturbed (Figure 9E–H). These results suggest that AP involvement in SNP-improved tolerance is not achieved by influencing the ascorbate-glutathione cycle.

### 3.9. Effects of Exogenous NO on Antioxidant Enzymes in the Presence of SHAM under Cd Stress

The antioxidant enzyme activities were enhanced under various stresses to reduce the level of ROS [36,42]. As shown in Figure 9A–D, under Cd stress, SOD activity was elevated by 28.6% and 35.8%, CAT activity by 50.4% and 56.7%, and POD activity by 1.5 times and 1.7 times in Ganpi6 and Kunlun14, respectively. Under Cd + SNP treatment, activities of all these enzymes (except POD) were further increased in Ganpi6 and Kunlun14 roots. Nevertheless, applying SHAM had nearly no distinct impact on these antioxidant enzymes under Cd + SNP treatment. These results suggest that AP involvement in SNP-improved Cd tolerance in Ganpi6 and Kunlun14 is not achieved by affecting the antioxidant enzyme system (Figure 9A–D).

## 4. Discussion

In this study, we examined the responses of barley (Ganpi6) and highland barley (Kunlun14) to Cd stress, and further explored the possible function of the AOX pathway (AP) in NO-enhanced tolerance to Cd stress. The results suggest that AP is involved in NO-enhanced tolerance to Cd stress in barley roots by controlling ROS accumulation.

Oxidative stress is considered as one of important responses to Cd toxicity [56,57]. In our experiment, we observed that the dry weight and the root growth were markedly decreased. However, EL and MDA were obviously increased in two varieties with increased Cd concentration (Figure 1). As a result, Cd stress induced serious oxidative damage in two varieties.

How Cd activates AP is largely unknown in plants. Recently, some reports suggest that NO could relieve heavy metal toxicity [14,15,51]. In this study, to explore the role of NO in regulating AP under Cd stress in two barley roots, NO donor (SNP) and scavenger (PTIO) were used. Results indicate that NO plays an important role in response to Cd stress. When AP was inhibited under Cd stress, NO fluorescence signal was enhanced (Figure 3), implying that NO and AP have a close relationship in response to Cd stress. Recently, research has reported that NO ameliorates arsenic toxicity by altering the alternative oxidase (*Aox1*) gene expression in *Hordeum vulgare* L [58]. Applying exogenous SNP relieved Cd-induced oxidative stress in Ganpi6 and Kunlun14 roots. However, when AP was inhibited by SHAM, Cd-induced oxidative stress was further aggravated and it was not reversible by application of SNP (Figure 7), suggesting AP plays an indispensable role in response to Cd stress and participates in the SNP-improved Cd stress tolerance. This is perhaps another crucial reason for the enhanced Cd tolerance in two barley roots. When AP was inhibited by SHAM under Cd stress, there was nearly no effect on Cd content in two barley shoots or roots (Appendix A), suggesting that AP is not involved in enhancing Cd tolerance by decreasing Cd uptake or increasing discharge of Cd. In addition, Cd content in two barley shoots was minor compared with Cd content in roots. Therefore, inhibition of Cd transport from root to stem could be one of the reasons for the tolerance of barley to Cd stress. We further observed that Cd-induced NO mainly resulted from the NR pathway, which is consistent with some earlier reports [59,60,61]. Some studies have demonstrated that NOS-regulated NO production is engaged in salt stress and zinc tolerance [21,62], and another report revealed that NO production results from both NR- and NOS-depended pathways [60]. The different source of NO production might be due to various stresses and plant species.

Antioxidant molecules and antioxidant defense system are core elements in keeping ROS homeostasis in barley plants exposed to various stressors [63,64,65]. Applying SNP could stimulate the antioxidant defense system to enable plant tolerance to stressors [37,66,67]. Substantial evidence showed that AOX functions in maintaining ROS homeostasis [33]. A lack of AOX results in elevated activity of ROS-related scavenging enzymes in some plants [48,68]. In this study, we found that the activities of ROS-related scavenging enzymes and antioxidant molecules were enhanced in two barley roots under normal and stressed conditions in the presence of SNP, which is in line with the previous study in barley seedlings [64]. The results also showed that SHAM had nearly no effects on ROS-related scavenging enzymes and antioxidant molecules (Figure 8 and Figure 9). Furthermore, the results showed that the correlation between AP and ROS-related scavenging enzymes or antioxidant molecules was minor (Appendix A), suggesting that AP engagement in SNP-elevated tolerance is not achieved by affecting ROS-related scavenging enzymes and antioxidant molecules. This observation is different from some previous studies [33,34], possibly due to different stress types and intensity. These results indicated that the elevated ROS-related scavenging enzymes and antioxidant molecules might also be in charge of NO-enhanced tolerance to Cd stress in two barley varieties.

## 5. Conclusions

On the basis of these results, a signal pathway model was proposed (Figure 10). It illustrates the physiological and biochemical mechanisms of the specific role of AOX pathway involvement in the NO-mediated Cd tolerance in two barley varieties. NO, which is mainly derived from the NR pathway, is stimulated in response to Cd stress. NO acts as a signal molecule to affect the expression of *HvAOX1a*, *HvAOX1d1*, and *HvAOX1d2*; corresponding AOX proteins; and ultimately V_alt_. NO inhibited the ROS burst by promoting AP and ROS-related scavenging enzymes and antioxidant molecules. There was no correlation between AP and antioxidant enzymes or the ascorbate-glutathione cycle in response to Cd stress. Heavy metal tolerance requires an integrated physiological and biochemical processes. In our study, we prove that the AOX pathway plays an indispensable role in the SNP-elevated resistance to Cd stress in barley roots. However, it is worth noting that the capacity of AP is not the actual activity of AP [69,70]. Measurements of the AP activity in vivo should be done in future because it is crucial for determining the role of AOX in plants.

## Figures and Tables

**Figure 1 plants-08-00557-f001:**
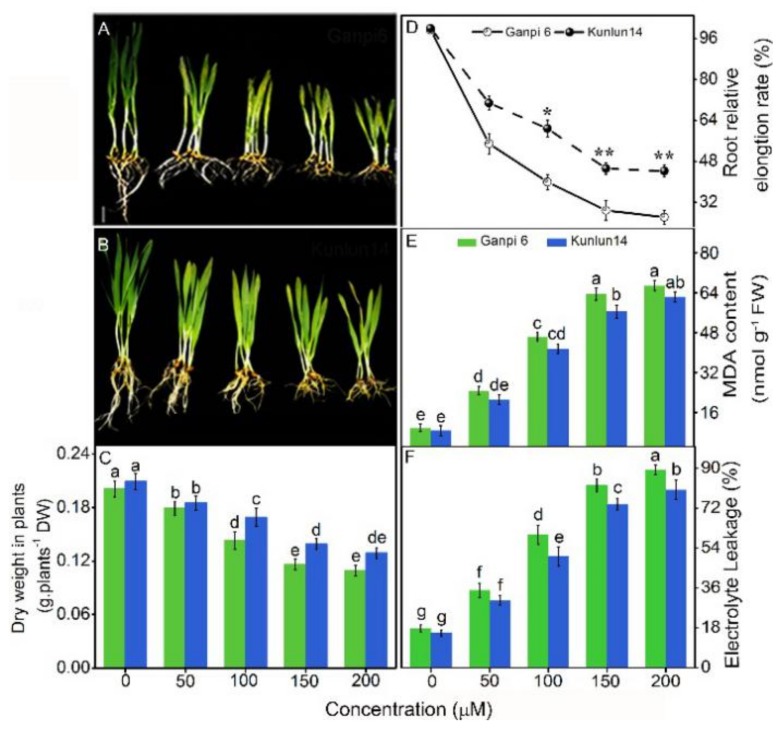
Effects of Cd on dry weight, root elongation, malondialdehyde (MDA), and electrolyte leakage (EL) in Ganpi6 and Kunlun14. (**A**) Growth of Ganpi6 seedlings under 0–200 µM Cd treatment. (**B**) Growth of Kunlun14 seedlings under 0–200 µM Cd treatment. (**C**) Dry weight of Ganpi6 and Kunlun14 seedlings. (**D**) Root relative elongation (RRE) of Ganpi6 and Kunlun14 seedlings (*n* > 20). (**E**) MDA contents in Ganpi6 and Kunlun14 roots under 0–200 µM Cd treatment. (**F**) EL in Ganpi6 and Kunlun14 roots under 0–200 µM Cd treatment. Bar in (**A**) = 1 cm. The six-day-old seedlings were grown in 1/4-strengh modified Johnson’s nutrient solution with 0–200 µM Cd for 48 h, and the primary root length, MDA contents, and EL were measured. Bars represent mean ± SE (*n* = 3). Different lower case letters represent significant difference at *p* < 0.05; * and ** represent difference at *p* < 0.05 and 0.01, respectively.

**Figure 2 plants-08-00557-f002:**
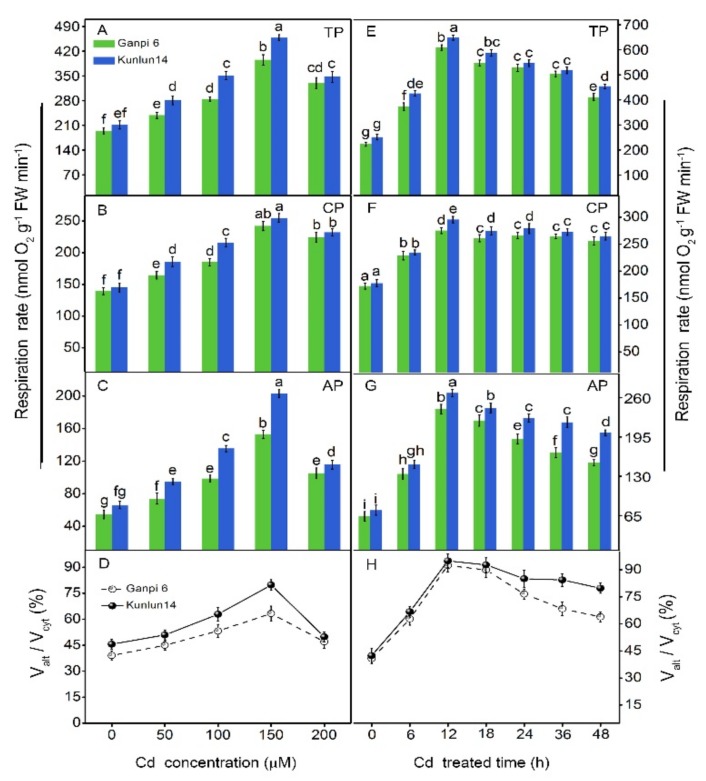
Effects of Cd on respiration rates in Ganpi6 and Kunlun14 roots. Changes of total respiration rate (TP) (**A**), cytochrome respiration capacity (CP) (**B**), alternative respiration capacity (AP) (**C**) and Valt/Vcyt (**D**) under 0–200 µM Cd treatment for 48 h. Changes of TP (**E**), CP (**F**), AP (**G**), and Valt/Vcyt (**H**) under 150 µM Cd treatment for different times. Ganpi6 and Kunlun14 seedlings were first grown in 1/4-strengh Johnson’s nutrient solution. After 6 days of growth, seedlings were treated with 0–200 µM Cd for 48 h. Bars represent mean ± SE (standard deviation) (n = 3) and different lower case letters represent significant difference at *p* < 0.05.

**Figure 3 plants-08-00557-f003:**
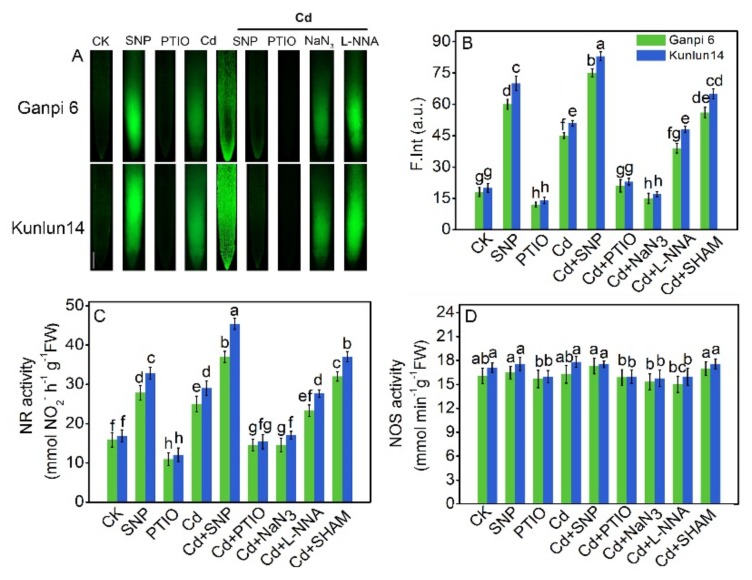
Effect of SNP (NO donor) on NO release. NO-dependent DAF-FM-DA (4-amino-5-methylamino-2’7’-difluorofluorescein diacetate) fluorescence staining in Ganpi6 and Kunlun14 roots (**A**), quantification of NO fluorescence intensity (F. Int.) in the images in arbitrary units (a.u.) (**B**), nitrate reductase (NR) (**C**), and nitric oxide synthase (NOS) (**D**) activities under Cd stress in Ganpi6 and Kunlun14 roots. Bar = 1 cm. In this experiment, 150 µM Cd, 150 µM SNP, 25 µM NaN3, 35 µM NOS inhibitor (L-NNA), 100 µM salicylhydroxamic acid (SHAM) were used. Bars represent mean ± SE (n = 3) and different lower case letters represent significant difference at *p* < 0.05.

**Figure 4 plants-08-00557-f004:**
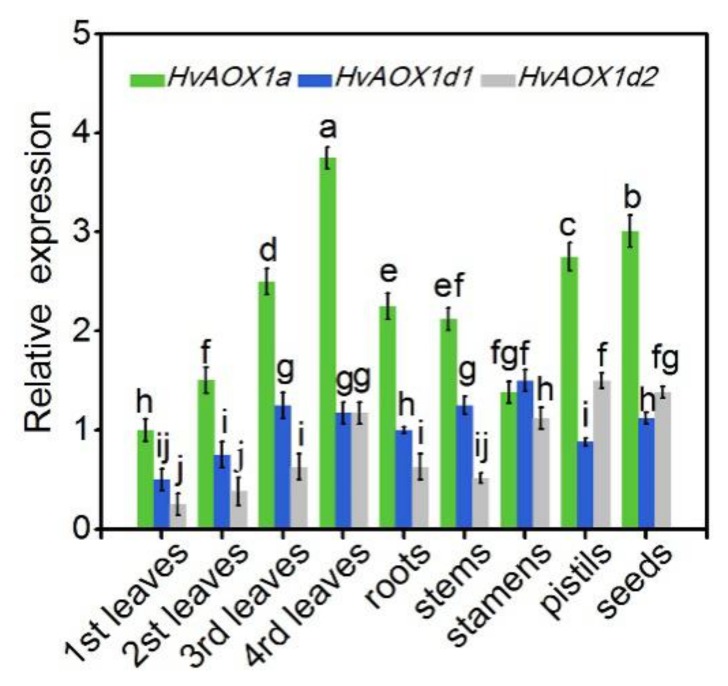
Relative expression of *HvAOXs* in Kunlun14. Different tissues of Kunlun14 were collected in the spring season. The relative transcript abundance was quantified by Rotor-Gene Real-Time Analysis Software 6.1. Bars represent mean ± SE (*n* = 3).

**Figure 5 plants-08-00557-f005:**
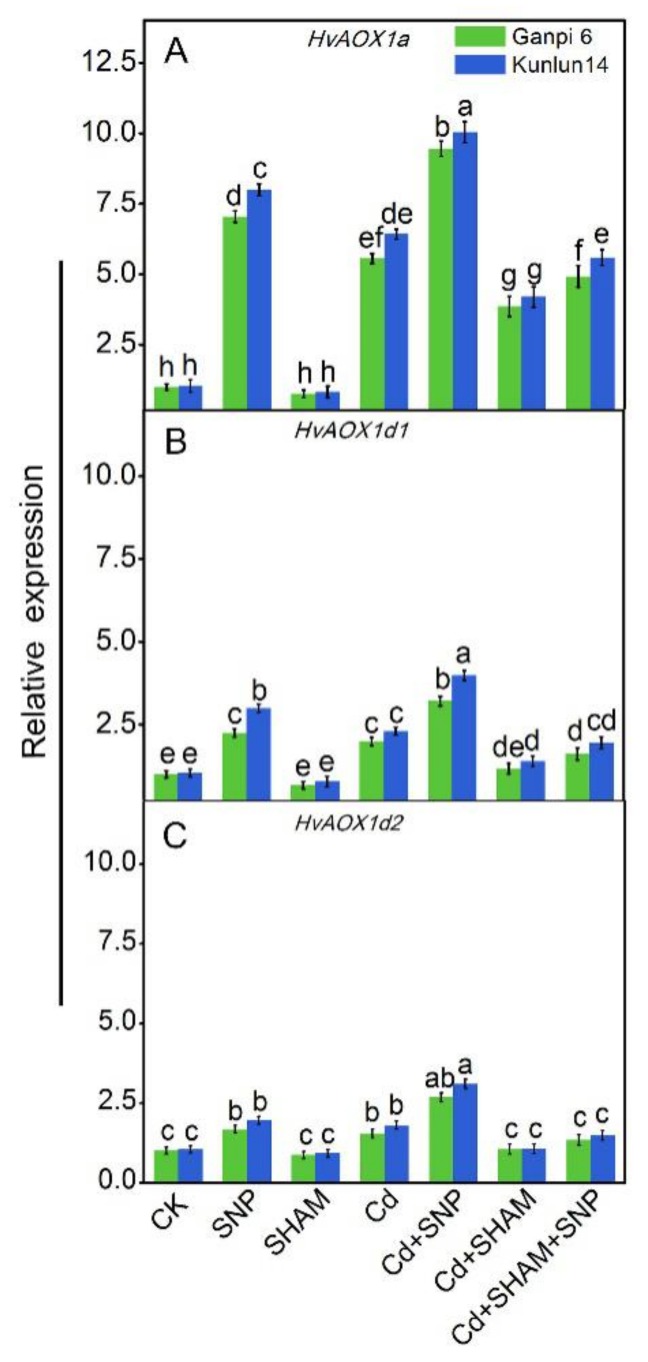
Effect of SNP and SHAM on the expression of *HvAOX* genes under Cd stress. *HvAOX1a* (**A**); *HvAOX1d1* (**B**); *HvAOX1d2* (**C**). In the experiment, 150 µM Cd, 150 µM SNP, and 100 µM SHAM were used. Bars represent mean ± SE (*n* = 3), and different lower case letters represent significant difference at *p* < 0.05.

**Figure 6 plants-08-00557-f006:**
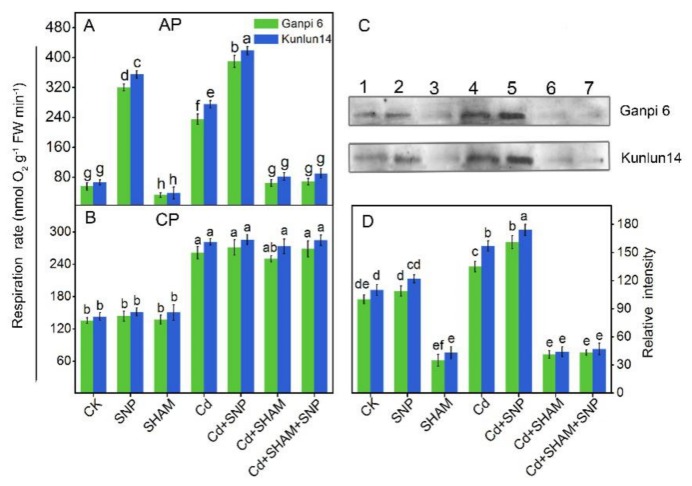
Effects of SNP on alternative pathway capacity (V_alt_) (**A**), cytochrome pathway capacity (V_cyt_) (**B**), western blot analysis (**C**), and quantification (**D**) of alternative oxidase (AOX) protein in the roots of Ganpi6 and Kunlun14 under Cd stress. In the experiment, 150 µM Cd, 150 µM SNP, and 100 µM SHAM were used. The lane numbers in (**C**) represent different treatments (Lane 1, CK; 2, SNP; 3, SHAM; 4, Cd; 5, Cd + SNP; 6, Cd + SHAM; 7, Cd + SHAM + SNP). Bars represent mean ± SE (*n* = 3), and different lower case letters represent significant difference at *p* < 0.05.

**Figure 7 plants-08-00557-f007:**
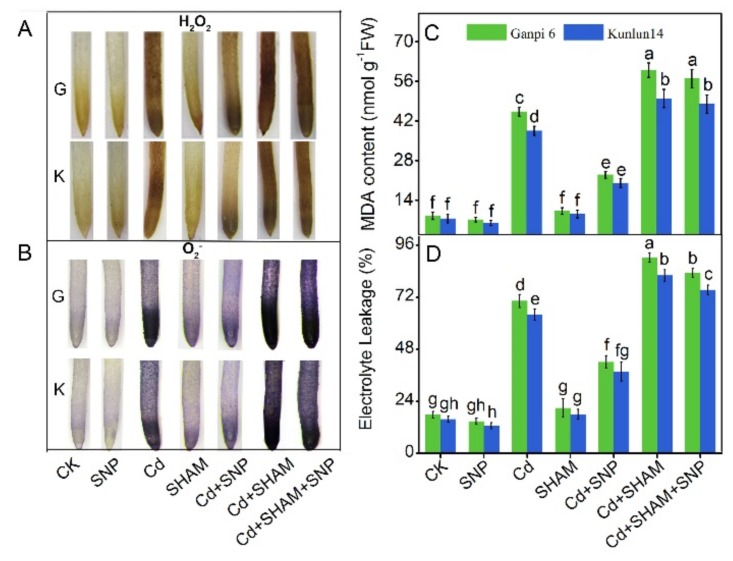
Effects of SNP and SHAM on H_2_O_2_ (**A**) and O_2_^−^ (**B**) accumulation and malonaldehyde (MDA) content (**C**) and electrolyte leakage (EL) (**D**) under Cd stress for 48 h in Ganpi6 and Kunlun14 roots. Bar = 0.5 cm. In the experiment, 150 µM Cd, 150 µM SNP, and 100 µM SHAM were used. CK represented the untreated roots of Ganpi6 and Kunlun14. Bars represent mean ± SE (*n* = 3) and different lower case letters represent significant difference at *p* < 0.05.

**Figure 8 plants-08-00557-f008:**
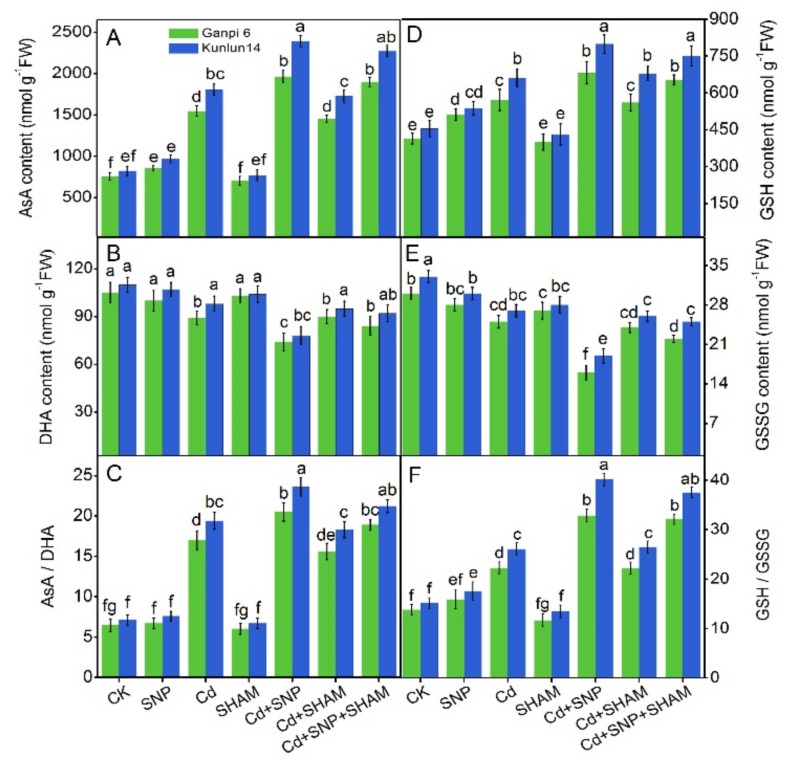
Effects of SNP and SHAM on ascorbic acid (AsA) (**A**), hydroascorbic acid (DHA) (**B**), AsA/DHA (**C**), reduced glutathione (GSH) (**D**), oxidized glutathione (GSSG) (**E**), and GSH/GSSG (**F**) under Cd stress for 48 h. In the experiment, 150 µM Cd, 150 µM SNP, and 100 µM SHAM were used. Bars represent mean ± SE (*n* = 3), and different lower case letters represent significant difference at *p* < 0.05.

**Figure 9 plants-08-00557-f009:**
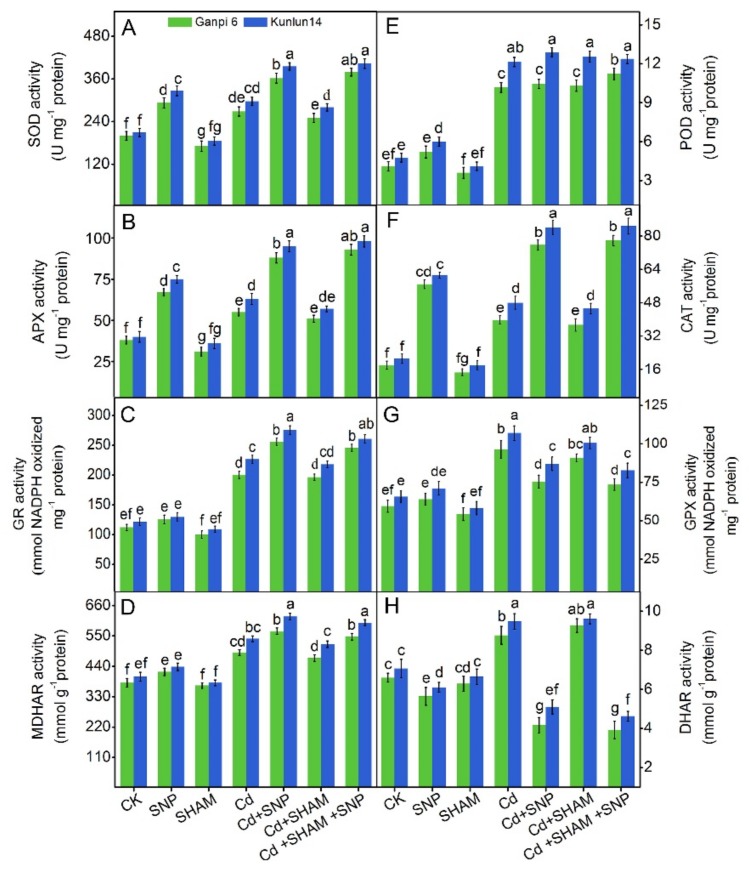
Effects of SNP and SHAM on the enzyme activities of (**A**) superoxide dismutase (SOD), (**B**) ascorbate peroxidase (APX), (**C**) glutathione reductase (GR), (**D**) monodehydroascorbic acid reductase (MDHAR), (**E**) peroxidase (POD), (**F**) catalase (CAT), (**G**) glutathione peroxidase (GPX), and (**H**) didehydroascorbic acid reductase (DHAR) under Cd stress for 48 h in Ganpi6 and Kunlun14 roots. In the experiment, 150 µM Cd, 150 µM SNP, and 100 µM SHAM were used. Bars represent mean ± SE (n = 3) and different lower case letters represent significant difference at *p* < 0.05.

**Figure 10 plants-08-00557-f010:**
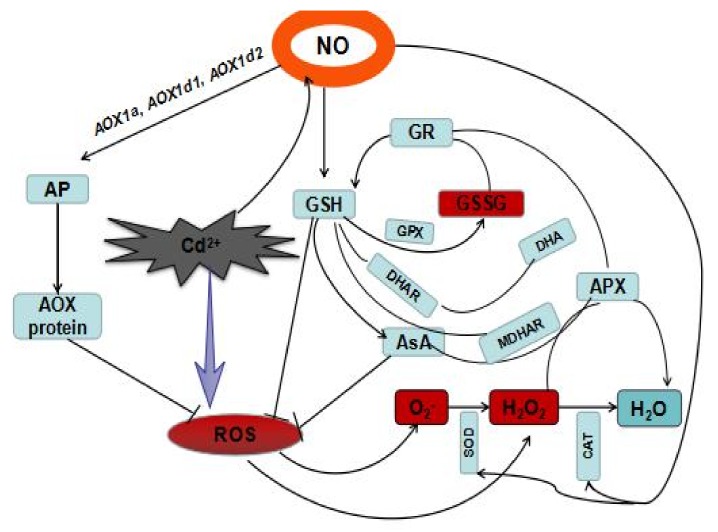
A diagram depicting the Cd-induced toxicity and the protective mechanism in barley roots. Arrows represent the enhanced effects and hyphens represent the suppressed effects.

**Table 1 plants-08-00557-t001:** Primer sequences.

Primer Name	Primer Sequence (5′—3′)
Hvactin-F	GTGGTCGTACAACWGGTATTGTG
Hvactin-R	GCTCATCAAATCCAAACACTG
ORFHvAOX1a-F	GCAACGAACCTACAAGCGTG
ORFHvAOX1a-R	GCTAAAGAGCCCTCATTTCCTC
ORFHvAOX1d1-F	CCTCCCATTAGCTTTTCGACCAG
ORFHvAOX1d1-R	CGGTAGCACGTAACAGCGTGGACT
ORFHvAOX1d2-F	TACGACCACGAGTTTCGCGAGCA
ORFHvAOX1d2-R	GCTAAAGAGCCCTCATTTCCTC
qHvAOX1a-F	GCAACGAACCTACAAGCGTG
qHvAOX1a-R	AAGAGCCCAGCACCAACAA
qHvAOX1d1-F	CCTCCCATTAGCTTTTCGACCAG
qHvAOX1d1-R	CGGTAGCACGTAACAGCGTGGACT
qHvAOX1d2-F	TACGACCACGAGTTTCGCGAGCA
qHvAOX1d2-R	GCTAAAGAGCCCTCATTTCCTC

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
