# Peer review of "Alternative Pathway is Involved in Nitric Oxide-Enhanced Tolerance to Cadmium Stress in Barley Roots"

_plants, 2019, doi:10.3390/plants8120557_

Round 1
Reviewer 1 Report
The presented work is partially innovative but deals with a very important topic in plant physiology which is still under investigation.
The proposed signal pathway model properly illustrates the physiological and biochemical mechanisms of the specific role of AOX pathway involvement in the NO-mediated Cd tolerance in highland barley. It also underlines that heavy metal tolerance requires an integrated physiological and biochemical processes.
I wonder how the authors got to the following conclusion?
"There is no correlation between AP and antioxidant enzymes or the ascorbate-glutathione cycle in response to Cd stress."
I suggest to do a correlation analysis between the all or most of studied parameters to show the specific mechanisms standing behind the response to cadmium (Cd) stress in 2 studied two barley varieties.
Author Response
Dear Reviewer,
Thank you so much for your constructive comments on our manuscript “Alternative pathway is involved in nitric oxide-enhanced tolerance to cadmium stress in barley roots” (Plants-626777). We have revised this manuscript according to the reviewers’ comments and highlighted all revisions in blue font. The English of the manuscript has been polished. I hope the revised version of the manuscript is satisfactory to you and the reviewers. In the point-by-point responses below, reviewers’ comments are in black fonts and our responses are in blue fonts.
Sincerely
Yurong Bi
Reviewer 1
The presented work is partially innovative but deals with a very important topic in plant physiology which is still under investigation. The proposed signal pathway model properly illustrates the physiological and biochemical mechanisms of the specific role of AOX pathway involvement in the NO-mediated Cd tolerance in highland barley. It also underlines that heavy metal tolerance requires an integrated physiological and biochemical processes.
>I wonder how the authors got to the following conclusion?
"There is no correlation between AP and antioxidant enzymes or the ascorbate-glutathione cycle in response to Cd stress."I suggest to do a correlation analysis between the all or most of studied parameters to show the specific mechanisms standing behind the response to cadmium (Cd) stress in 2 studied two barley varieties.
> Thanks for your constructive advice. In the revised MS, we analyzed the correlation between AP and enzymatic activities, ASC, GSH, NR and NiR activities (Suppl. Table.1). The results showed that the correlation between AP and ROS-related scavenging enzymes or antioxidant molecules was minor. The related content has been added to the Discussion in the revised manuscript (P15, L13-14).
Reviewer 2 Report
General scientific comments
In my opinion, this study concerns the mechanisms of enhancing tolerance to Cd stress in barley and not just in highland barley and both genotypes under study exhibit basically the same response. Varietal differences in Cd tolerance and accumulation in barley genotypes have been established in several studies (see specific comments). However, the data shown in the present study does not clearly support a differential in Cd tolerance between Kunlun14 and Ganpi6 and the literature on those varieties is scarce and doesn´t mention this trait. Moreover, taken in account the variability of genotypic responses to the trait, the comparison of only two genotypes is not enough to support the claim that “AOX pathway plays an indispensable role in the SNP-elevated resistance to Cd stress in highland barley”. I suggest re-writing the manuscript with less focus on establishing the mechanisms of tolerance in highland barley. The study on mechanistic differences between highland barley and conventional barley should be done but needs to include several genotypes of both types.
Another important point the authors should take into consideration is that enhanced tolerance to Cd stress in a crop is not necessarily a desirable trait if Cd is accumulated in the edible part.
I believe the discussion needs improvement with more comparison with previous studies on barley and even other crops. I suggest including the following studies in the discussion:
Chen et al. / J Plant Growth Regul (2010) 29:394–408
Wang et al. /Ecotoxicology and Environmental Safety, 135(2017)75–81
Peksen and Sanal / Fresenius Environmental Bulletin, 27(9) (2018) 5871-5881
Shukla et al. / EcotoxicologyandEnvironmentalSafety120(2015)59–65
Wang et al. / Scientific Reports 7: 14233, DOI:10.1038/s41598-017-14601-8
As claimed by the authors, there are only few studies on the topic of the mechanisms of enhanced resistance to Cd stress in barley, therefore this study is of relevance and of general interest to the scientific community. Moreover, the study is thorough, the experiments are described with enough detail and the data supports the involvement of the AP pathway in NO-enhanced tolerance.
Specific scientific comments/suggestions
The differential in Cd tolerance between the two barley genotypes is not evident. Was there a differential decrease in biomass? In figure 1 it does seem genotypes had different biomass and if measured should be included. Did the appearance and severity of Cd toxicity symptoms differed significantly between genotypes? (see Chen et al. / J Plant Growth Regul (2010) 29:394–408)
It would be important to determine the concentration of Cd in the root material (if there was a pre-treatment for removing adsorbed Cd) or in the leaf material. The question is, did the mechanism described also decrease the uptake of Cd (see Wang et al. /Ecotoxicology and Environmental Safety, 135(2017)75–81)
Other details:
Extensive revision of the text should be made. Below some examples.
Page 1
Line 3 –revise phrase starting with “It has become …”. I don´t understand the meaning
Line 7 – remove in plants
Page 2
2nd paragraph
Replace lessen by decrease
Replace was widely chosen to imitate by has been widely used to mimic
Replace to study various biological functions of NO in plants by in the study of its various biological functions in plants
Replace which might be an upstream signal molecule by which might functioned as an upstream signal molecule
Page 3
2.3 section
Replace on the basis of the method of [41] by on the basis of the method described in reference [41] or even better on the basis of the method described by Tang et al. (2014) [41]. (the same should be made throughout the materials section)
What is soft water?
Replace conducted by measured or determined (and throughout the manuscript)
Define normal temperature
2.4 section
Remove respectively
Figure 1
What is the meaning of one start and 2 stars in fig. 1C
Author Response
Dear Reviewer,
Thank you so much for your constructive comments on our manuscript “Alternative pathway is involved in nitric oxide-enhanced tolerance to cadmium stress in barley roots” (Plants-626777). We have revised this manuscript according to your comments and highlighted the revisions in blue font. The English of the manuscript has been polished. I hope the revised version of the manuscript is satisfactory to you. In the point-by-point responses below, reviewers’ comments are in black fonts and our responses are in blue fonts.
Sincerely
Yurong Bi
Reviewer 2
Comments and Suggestions for Authors
General scientific comments
>1. In my opinion, this study concerns the mechanisms of enhancing tolerance to Cd stress in barley and not just in highland barley and both genotypes under study exhibit basically the same response. Varietal differences in Cd tolerance and accumulation in barley genotypes have been established in several studies (see specific comments). However, the data shown in the present study does not clearly support a differential in Cd tolerance between Kunlun14 and Ganpi6 and the literature on those varieties is scarce and doesn´t mention this trait. Moreover, taken in account the variability of genotypic responses to the trait, the comparison of only two genotypes is not enough to support the claim that “AOX pathway plays an indispensable role in the SNP-elevated resistance to Cd stress in highland barley”. I suggest re-writing the manuscript with less focus on establishing the mechanisms of tolerance in highland barley. The study on mechanistic differences between highland barley and conventional barley should be done but needs to include several genotypes of both types.
> Thanks for your constructive advice. In the revised MS, the related description has been revised in the full text.
>2. Another important point the authors should take into consideration is that enhanced tolerance to Cd stress in a crop is not necessarily a desirable trait if Cd is accumulated in the edible part. I believe the discussion needs improvement with more comparison with previous studies on barley and even other crops. I suggest including the following studies in the discussion.
> Thanks for your constructive advice. In the revised MS, we added the related description in the Discussion, and the references were cited (P14, L32-33, 38-44; P15, L4,6).
>3. As claimed by the authors, there are only few studies on the topic of the mechanisms of enhanced resistance to Cd stress in barley, therefore this study is of relevance and of general interest to the scientific community. Moreover, the study is thorough, the experiments are described with enough detail and the data supports the involvement of the AP pathway in NO-enhanced tolerance.
>Thank you for the positive comments on this study.
Specific scientific comments/suggestions
>4. The differential in Cd tolerance between the two barley genotypes is not evident. Was there a differential decrease in biomass? In figure 1 it does seem genotypes had different biomass and if measured should be included. Did the appearance and severity of Cd toxicity symptoms differed significantly between genotypes? (see Chen et al. / J Plant Growth Regul (2010) 29:394–408)
> In the revised MS, the changes of dry weight (Fig.1C) was added, and the results showed that the biomass decreased more in Ganpi-6 than that in Kunlun14. Cd stress induced more severely damages in Ganpi-6 than in Kunlun-14, such as higher accumulation of ROS (O2-, H2O2), elevated MDA and EL levels.
>5. It would be important to determine the concentration of Cd in the root material (if there was a pre-treatment for removing adsorbed Cd) or in the leaf material. The question is, did the mechanism described also decrease the uptake of Cd (see Wang et al./Ecotoxicology and Environmental Safety, 135(2017)75–81)
> The content of Cd in roots and leaves has been detected based on the method described in reference [66], and in the revised MS, we added the data in the supplemental materials. The results showed that when AP was inhibited by SHAM under Cd stress, there was nearly no effect on Cd content in roots and shoots of both barley varieties compared to Cd treatment alone, and the content was added in the Discussion (P14, L39-44)
>The method:
>Cd content was assayed based on the method described in reference [66]. Barley shoots and roots were washed with the deionized water for three times, then dried at 80 °C in oven to a constant weight, 0.1 g dried tissues was mixed with HNO3 and digested in a microwave digestion instrument for 4h. Then the solution was collected to measure the Cd content with ICP-MS.
Other details:
Extensive revision of the text should be made. Below some examples.
>Page 1
Line 3 –revise phrase starting with “It has become …”. I don´t understand the meaning
> In the revised MS, the sentence has been deleted.
Line 7 – remove in plants
> Thanks. It has been removed.
>Page 2
2nd paragraph
Replace lessen by decrease
> It has been revised (P2;L28).
Replace was widely chosen to imitate by has been widely used to mimic
> It has been revised (P2;L33).
Replace to study various biological functions of NO in plants by in the study of its various biological functions in plants
> It has been revised (P2;L34).
Replace which might be an upstream signal molecule by which might functioned as an upstream signal molecule
> Thanks. It has been revised (P2;L37).
>Page 3
2.3 section
Replace on the basis of the method of [41] by on the basis of the method described in reference [41] or even better on the basis of the method described by Tang et al. (2014) [41]. (the same should be made throughout the materials section)
> Thanks for your constructive advice. The related contents have been revised (P3;L22,34-35,42; LP4;L3,11,23).
What is soft water?
> The soft water is the deionized water, and it has been revised (P6;L26-27).
Replace conducted by measured or determined (and throughout the manuscript)
> Thanks for your constructive advice. The related contents have been revised (P3;L22; LP4;L11,23).
Define normal temperature
> The normal temperature is room temperature, and it has been revised (P3;L30).
>2.4 section
Remove respectively
> It has been deleted.
Figure 1
What is the meaning of one start and 2 stars in fig. 1C
> * and ** represent P<0.05 and 0.01, respectively. And it has been revised (P7;L3).

Reviewer 3 Report
The manuscript ‘ID plants-626777’ investigates the role of AP in response to cadmium (Cd) stress in two barley varieties, highland barley (Kunlun14) and barley (Ganpi6). They observed that malondialdehyde (MDA) content and electrolyte leakage (EL) level under Cd stress increased more in Ganpi6 than in Kunlun14. The expressions of alternative oxidase (AOX) genes (mainly AOX1a), AP capacity (Valt) and AOX protein amount were also induced more in Kunlun14 under Cd stress. Moreover, H2O2 and O2- contents were raised more in Ganpi6 than in Kunlun14 Cd-treated roots, suggesting that AP contributed to NO-enhanced Cd stress tolerance. These observations lead authors to strongly conclude that AP exerted an indispensable function in NO-enhanced Cd stress tolerance in Kunlun14. I have reviewed the manuscript and based on my own reading, the manuscript is novel, interesting, and well presented. Although I consider it is not suitable for publication in Plants in the present form, it might become suitable following minor revision concerning terms AP capacity and activity. On one hand, the english writting needs some imporvement, I recomend the help of a native speaker. On the other hand, the authors mostly ignored research conducted years ago in relation to the model that explains how respiration and electron partitioning between cytochrome and alternative oxidase pathways actually function (Ribas-Carbo et al 1995; and for review see Day et al., 1996 or Millenaar and Lambers, 2003). This is still the model that is accepted by the scientific community in the AOX research field (see review of Del-Saz et al 2018). It has been widely demonstrated that in vivo activity measurements are crucial for determining the role of AOX in plants. While I would not ask for this data, as I am well aware it involves a very difficult technique that is not readily available, I think this issue must be discussed and suggestions for future work should be made. Authors have measured AP capacity and the manuscript reflect it as such. However they made no distinction between terms AP capacity and AP activity along the manuscript, which may led to create confusion to the reader as AP activity does not always correlates with its capacity.
Author Response
Dear Reviewer,
Thank you so much for your constructive comments on our manuscript “Alternative pathway is involved in nitric oxide-enhanced tolerance to cadmium stress in barley roots” (Plants-626777). We have revised this manuscript according to your comments and highlighted the revisions in blue font. The English of the manuscript has been polished. I hope the revised version of the manuscript is satisfactory to you. In the point-by-point responses below, reviewers’ comments are in black fonts and our responses are in blue fonts.
Sincerely
Yurong Bi
Reviewer 3
The manuscript ‘ID plants-626777’ investigates the role of AP in response to cadmium (Cd) stress in two barley varieties, highland barley (Kunlun14) and barley (Ganpi6). They observed that malondialdehyde (MDA) content and electrolyte leakage (EL) level under Cd stress increased more in Ganpi6 than in Kunlun14. The expressions of alternative oxidase (AOX) genes (mainly AOX1a), AP capacity (Valt) and AOX protein amount were also induced more in Kunlun14 under Cd stress. Moreover, H2O2 and O2- contents were raised more in Ganpi6 than in Kunlun14 Cd-treated roots, suggesting that AP contributed to NO-enhanced Cd stress tolerance.
These observations lead authors to strongly conclude that AP exerted an indispensable function in NO-enhanced Cd stress tolerance in Kunlun14. I have reviewed the manuscript and based on my own reading, the manuscript is novel, interesting, and well presented.
Although I consider it is not suitable for publication in Plants in the present form, it might become suitable following minor revision concerning terms AP capacity and activity. The authors mostly ignored research conducted years ago in relation to the model that explains how respiration and electron partitioning between cytochrome and alternative oxidase pathways actually function (Ribas-Carbo et al 1995; and for review see Day et al., 1996 or Millenaar and Lambers, 2003).
This is still the model that is accepted by the scientific community in the AOX research field (see review of Del-Saz et al 2018). It has been widely demonstrated that in vivo activity measurements are crucial for determining the role of AOX in plants.
While I would not ask for this data, as I am well aware it involves a very difficult technique that is not readily available, I think this issue must be discussed and suggestions for future work should be made.
Authors have measured AP capacity and the manuscript reflect it as such. However they made no distinction between terms AP capacity and AP activity along the manuscript, which may led to create confusion to the reader as AP activity does not always correlates with its capacity.
> Thanks for your constructive advice. In the revised MS, the related content has been added (P15; L32-35)
Round 2
Reviewer 2 Report
I appreciate that the authors have taken my suggestions into consideration and I am happy with the information added. In my opinion the manuscript should be published in the present form.